# The Antagonist Effect of Arachidonic Acid on *GLUT4* Gene Expression by Nuclear Receptor Type II Regulation

**DOI:** 10.3390/ijms20040963

**Published:** 2019-02-22

**Authors:** Inmaculada Moreno-Santos, Sara Garcia-Serrano, Hatim Boughanem, Lourdes Garrido-Sanchez, Francisco José Tinahones, Eduardo Garcia-Fuentes, Manuel Macias-Gonzalez

**Affiliations:** 1Department of Endocrinology and Nutrition, Virgen de la Victoria University Hospital, University of Malaga (IBIMA), 29010 Malaga, Spain; morenosantos.inma@gmail.com (I.M.-S.); lourgarrido@gmail.com (L.G.-S.); fjtinahones@hotmail.com (F.J.T.); 2CIBER de Diabetes y Enfermedades Metabólicas Asociadas (CIBERDEM), Unidad de Gestión Clínica de Endocrinología y Nutrición, Instituto de Investigación Biomédica de Málaga (IBIMA), Hospital Regional Universitario, 29010 Málaga, Spain; garciasara79@hotmail.com; 3Instituto de Investigación Biomédica de Málaga (IBIMA), Facultad de Ciencias, Universidad de Málaga, 29010 Málaga, Spain; h.b.boughanem@gmail.com; 4CIBEROBN (CIBER in Physiopathology of Obesity and Nutrition CB06/03/0018), Instituto de Salud Carlos III, 28029 Madrid, Spain; 5Department of Gastroenterology, Virgen de la Victoria University Hospital, Instituto de Investigación Biomédica de Málaga (IBIMA), University of Malaga, 29010 Malaga, Spain

**Keywords:** obesity, insulin resistance, LXR-α, GLUT4, fatty acids, PPAR-γ2, arachidonic acid

## Abstract

Objectives: Obesity is a complex disease that has a strong association with diet and lifestyle. Dietary factors can influence the expression of key genes connected to insulin resistance, lipid metabolism, and adipose tissue composition. In this study, our objective was to determine gene expression and fatty acid (FA) profiles in visceral adipose tissue (VAT) from lean and morbidly obese individuals. We also aimed to study the agonist effect of dietary factors on glucose metabolism. Design and methods: Lean and low and high insulin resistance morbidly obese subjects (LIR-MO and HIR-MO) were included in this study. The gene expression of liver X receptor type alpha (LXR-α) and glucose transporter type 4 (GLUT4) and the FA profiles in VAT were determined. Additionally, the in vivo and in vitro agonist effects of oleic acid (OA), linoleic acid (LA), and arachidonic acid (AA) by peroxisome proliferator-activated receptor type gamma 2 (PPAR-γ2) on the activity of GLUT4 were studied. Results: Our results showed a dysregulation of GLUT4 and LXR-α in VAT of morbidly obese subjects. In addition, a specific FA profile for morbidly obese individuals was found. Finally, AA was an PPAR-γ2 agonist that activates the expression of GLUT4. Conclusions: Our study suggests a dysregulation of LXR-α and GLUT4 expression in VAT of morbidly obese individuals. FA profiles in VAT could elucidate their possible role in lipolysis and adipogenesis. Finally, AA binds to PPAR-γ2 to activate the expression of GLUT4 in the HepG2 cell line, showing an alternative insulin-independent activation of GLUT4.

## 1. Introduction

Obesity is a complex but preventable disease that has a great impact on public health. Its prevalence is increasing and already extends worldwide. The impact of obesity is denoted by the fact that it is considered to be a significant risk factor for nearly every chronic condition, from impaired glucose tolerance to insulin resistance to metabolic syndrome [1]. More specifically, excess visceral fat (also referred to as central obesity) has a strong association with obesity-related comorbidities [2]. 

Diet, as a part of lifestyle modification, is the first strategy for the prevention and treatment of obesity. Nutrients usually interact with several nutrient-sensitive transcription factors to regulate gene expression and metabolic response [3,4]. However, the mechanisms of nutrient-dependent interaction at the genetic, molecular level and in cell homeostasis are not fully understood.

Recently, different transcriptional factors have been identified to be important regulators of glucose and lipid metabolism. Nuclear receptors (NRs) are a family of ligand-regulated transcription factors that directly regulate genes, which control a wide variety of biological processes. Among them, the retinoid X receptor (RXR), as a NR, acts as a sensor of multiple nutrients and leads the principal responses for metabolic processes [5]. Liver X receptor type alpha (LXR-α), an oxysterol-regulated nuclear receptor, also plays a complex role in cholesterol, glucose, and lipid homeostasis [6]. These receptors function by forming a heterodimer. Then, they interact with DNA at specific sites, known as liver X receptor response elements (LXREs). All such processes are highly regulated by a complex interaction between ligands and co-factors [5,7]. 

The role of LXR-α in adipogenesis is still controversial. LXR-α is expressed in mature murine and human adipocytes and is upregulated during fat cell differentiation [8]. It has been reported that peroxisome proliferator-activated receptor type gamma (PPAR-γ) and CCAAT-enhancer-binding protein type alpha (C/EBP-α) regulate the expression of LXRs in murine 3T3-L1 cells and human in vitro-differentiated Simpson–Golabi–Behmel syndrome pre-adipocytes [9,10]. However, the activation of LXRs promotes an improvement in adipogenesis when PPAR-γ is partially activated. The full activation of PPAR-γ suppresses the effect of LXR on adipocyte differentiation. Thus, the function of LXR in adipose tissue could be as a modulator of adipogenesis [11]. In preadipocytes, LXR-α is related to the regulation of lipogenesis and adipocyte-specific gene expression, although those studies were limited and inconsistent [12].

The effect of LXR-α agonists on adipogenesis is still a matter of debate. While in vivo and in vitro studies in adipose cells found that LXR-α can be activated by the endogenous and synthetic LXR-α agonists, both have been shown to induce the LXR-α responsive gene in adipocytes [13]. 

LXR has also been implicated in glucose homeostasis, although additional studies will be required to better define its specific role. A study carried out on 3T3-L1 cells stimulated by LXR-α agonist T0901317 led to overexpression of LXR-α, increased basal glucose uptake, and glycogen synthesis [11]. In contrast, another study conducted on the same cell line determined that the glucose transporter type 4 (GLUT4) gene was regulated by LXR agonist GW3965 in vitro and murine adipose tissue in vivo. It was also reported that the LXR-α/RXR dimer binds to murine and human functional LXREs and regulates the activity of the reporter constructs driven by the GLUT4 promoter [10,14]. However, several subsequent studies have not been able to confirm these results [6]. Therefore, all this evidence suggests that LXR-α has a complex effect on the expression of glucose transporters. Moreover, the role of LXR on its modulation remains to be established.

Similarly, PPAR-γ2 is another ligand-activated transcription factor that belongs to the steroid hormone receptor superfamily [8]. Physiologically, both LXR-α and PPAR-γ2 need to form heterodimers with RXR to regulate the expression of their target genes [15,16]. Thus, a tight crosstalk should link LXR-α to PPAR-γ2 [17,18]. 

PPAR-γ2 is exclusively expressed in adipose tissue, which explains why it is required for adipogenesis and insulin sensitivity processes [19,20]. PPAR-γ is the main regulator of adipogenesis [9,21]. Despite the known biological role of PPAR-γ in adipogenesis, this process is multifactorial, and multiple regulatory proteins are involved. In the final stages of adipogenesis, PPAR-γ activates the expression of C/EBP-α and vice versa [22]. Therefore, there is a positive feedback loop between PPAR-γ and C/EBP-α [23,24].

PPAR-γ2 has also been implicated in important metabolic disorders, such as obesity and insulin resistance. Interestingly, numerous PPAR-γ2 agonists have been designed to treat these disorders [25,26,27]. 

Previous studies have found that PPAR-γ directly regulates the expression of GLUT4 [28]. A study reported that treatment with PPAR-γ agonists, such as thiazolidinediones (TZD), increase the expression of the insulin-dependent *GLUT4* gene. However, in another study in human adipocytes, no effect was detected [29]. In animal models of obesity and diabetes, in which expression of the GLUT4 was reduced in adipocytes, treatment with troglitazone (PPAR-γ agonist) corrects GLUT4 expression [30,31]. Interestingly, another study carried out in adipocytes 3T3-L1 pointed out that the experimental deletion of PPAR-γ decreases the transport of glucose stimulated by insulin, and this is due to a decrease in the GLUT4 function [32]. However, the antidiabetic effects of troglitazone and rosiglitazone, considered potent PPAR-γ agonists, have shown conflicting results as hypoglycemic agents regarding *GLUT4* gene regulation [27]. 

Numerous studies have investigated endogenous and exogenous PPAR-γ ligands. The exogenous agonists of PPAR-γ promote adipocytes differentiation and have antidiabetic properties [33]. Glitazone or TZD (pioglitazone and rosiglitazone) are used clinically as insulin sensitizers [34]. In addition, LXR-α and PPAR-γ bind to a number of metabolic intermediates, such as fatty acids (FA), as endogenous ligands and may function as metabolic sensors. Polyunsaturated FA (PUFA) of the n-3 and n-6 family, such as arachidonic acid (AA), eicosapentaenoic (EPA), docosahexaenoic (DHA) and linoleic acid (LA), are competitive antagonists of the interaction between LXR, PPARs, and their ligands [17,18]. Indeed, endogenous PUFAs can activate PPAR-γ [35]. However, more studies are needed to clarify the metabolic action pathways.

With these antecedents, it is clear that further studies are needed to understand in depth the implication of visceral adipose tissue (VAT) on the gene–diet interaction. Our aim of this study is to determine how the expression of key genes in the glucose and lipid metabolism could have an effect on adipogenesis. The composition of FAs in VAT may help to decipher the specific profiles of FAs in insulin-dependent obese individuals and could also evaluate the agonist effect of FAs and find new therapeutic options to treat insulin-dependent obesity.

## 2. Results

### 2.1. Anthropometric and Biochemical Parameters of Lean and Morbidly Obese Individuals

The anthropometric and biochemical parameters of the low insulin resistance morbidly obese (LIR-MO) and high insulin resistance morbidly obese (HIR-MO) individuals and lean individuals are presented in Table 1. There were no differences in age or gender among the groups. The morbidly obese groups presented significantly higher levels of insulin, free fatty acids (FFA), triglycerides, leptin, adiponectin, and homeostasis model assessment of insulin resistance (HOMA-IR) results (*p* < 0.05). Within the morbidly obese group, the HIR-MO subjects showed increased levels of glucose, insulin, triglycerides, and HOMA-IR in comparison with the LIR-MO individuals (*p* < 0.05). 

### 2.2. GLUT4 and LXR-α are Downregulated in Visceral Adipose Tissue in Morbidly Obese Individuals

Based on the role of LXR-α and GLUT4 in lipid metabolism, we investigated their expression pattern in VAT. RT-PCR analysis showed that GLUT4 expression was significantly decreased in VAT of morbidly obese subjects (*p* < 0.05). In addition, LIR-MO individuals had lower GLUT4 expression than that of the HIR-MO groups (*p* < 0.05) (Figure 1A). We confirmed the GLUT4 mRNA expression by western blot analysis using specific antibodies for the GLUT4 protein. This analysis corroborates the presence of the GLUT4 protein in VAT of the lean and morbidly obese individuals. However, the protein levels of GLUT4 were higher in the healthy subjects, while the LIR-MO group showed the lowest protein levels (Figure 1A).

On the other hand, RT-PCR analysis showed significantly increased LXR-α expression in VAT of lean subjects (*p* < 0.05). In addition, LXR-α expression of the LIR-MO subjects was significantly increased in comparison with that of the HIR-MO individuals (*p* < 0.05) (Figure 1B). Western blot analysis confirmed these results. The LXR-α protein was present in VAT of all of the groups. The lean subjects showed the highest protein levels, while the HIR-MO subjects showed the lowest protein levels.

### 2.3. The Fatty Acid Profile of the Visceral Adipose Tissue in the Non-Obese and Morbidly Obese Subjects

We further analyzed the profile of the FAs in VAT from the lean and obese morbidly subjects (Table 2). The morbidly obese individuals had a significantly higher concentration of palmitic acid (16:0), palmitoleic acid (16.1:n7), adrenic acid (22:4n6), PUFA, omega-6, and the saturated fatty acid (SFA)/PUFA ratio and a lower concentration of arachidic acid (20:0) (*p* < 0.05). Within the morbidly obese groups, the HIR-MO subjects showed decreased concentrations of lauric acid (12:0), myristic acid (14:0), DHA (22:6n3), the omega-3/omega-6 ratio and an increased concentration of arachidic acid (20:0), eicosenoic acid (20:1n9), AA (20:4n6), and the AA/LA ratio in comparison with the LIR-MO subjects (*p* < 0.05). 

### 2.4. Arachidonic Acid Binds to PPAR-γ2 and Their Ligands’ Actions

Oleic acid (OA) and LA are the most common FAs in the Mediterranean diet. AA also appears to have a sufficient impact in the Mediterranean diet, because it is synthesized from LA. It is, therefore, very interesting to consider these three FAs as possible ligands, to better understand the gene–diet interaction in obesity. For this reason, we aimed to investigate whether OA, LA, and AA bind directly to PPAR-γ2. A glutathione S-transferase (GST) pull-down assay was performed using an in vitro translated [^35^S]-labeled wild-type PPAR-γ2 (Figure 2A). The proteins were incubated in the presence of 1 μM of OA, LA, and AA and dimethyl sulfoxide (DMSO) as a vehicle. As a positive control, we used rosiglitazone (1 µM). Here, PPAR-γ2 did not show any association with human steroid receptor co-activator 1 (SRC1) in the presence of OA and LA. However, PPAR-γ2 showed reasonable levels of association with SRC1 in the presence of AA, which was significantly increased (*p* < 0.05).

To further support these results, we performed reporter gene assays from the extracts of HepG2 (liver hepatocellular carcinoma cell line) cells that were transiently transfected with a luciferase reporter gene construct under the control of four copies of human GLUT4-RE (glucose transporter type 4) fused with the thymidine kinase (*tk*) promoter. HepG2 cells transfected with PPAR-γ2, were treated with different FAs and synthetic agonists in the presence of the specific coactivator SCR1. In this assay, we observed that stimulation with 10 μM of different compounds caused different changes in the basal activity compared with the vehicle treatment. However, the overexpression of SRC1 caused further induction of the basal activity of the PPAR-γ2–RXRα interaction on GLUT4 and was only significantly increased by the AA and rosiglitazone (positive control) compared with the vehicle treatment (*p* < 0.05). No significant difference was noted in the basal activity in the presence of OA and LA (Figure 2B). We also checked the basal activity with other fatty acids, such as palmitic and palmitoleic acids, as shown in Table 2, and we did not observe any effect (data not shown). 

### 2.5. Arachidonic Acid Mediates Repression Actions of GLUT4 mRNA Expression via PPAR-γ2 Activation

To confirm the agonist effect of the AA on PPAR-γ2, the mRNA expression of GLUT4 was examined in HepG2 cells. The cells were transfected with PPAR-γ2-specific siRNA and incubated for 48 h in the presence of different compounds. After that, while the expression of GLUT4 was significantly increased in the presence of AA and the absence of PPAR-γ2-specific siRNA, no effect was observed in the GLUT4 expression when we silenced PPAR-γ2 through PPAR-γ2-specific siRNA transfect (Figure 3). Therefore, AA was able to specifically activate GLUT4 mRNA expression via PPAR-γ2.

## 3. Discussion

In the present study, we investigated the expression of LXR-α and GLUT4 in VAT from morbidly obese and lean subjects. This analysis showed decreased levels of GLUT4 and LXR-α in morbidly obese individuals. This finding could indicate a dysregulated GLUT4 and LXR-α pathway related to morbid obesity, suggesting a possible link to insulin-mediated glucose uptake in adipose tissue. In fact, the expression of GLUT4 in adipocytes is dependent on LXR-α [14]. The simultaneous decrease in the expression of GLUT4 and LXR-α leads us to conclude that one of the different pathways of obesity-related insulin resistance has taken place, in part, in VAT. Although changes in gene expression in adipose tissue and VAT alone are not sufficient to cause insulin resistance, alterations in the muscles, liver, and pancreatic β cells are required [36,37]. However, recent studies in mice without adipose tissue, found typical hallmarks of insulin resistance [38]. 

In addition, within the morbidly obese groups, we also found that the GLUT4 expression was significantly more increased in the HIR-MO subjects than in the LIR-MO subjects. Insulin has a regulatory effect on GLUT4 expression. A study carried out by insulin treatment in a mature 3T3-F442A adipocytes cell line in a medium containing glucose reported an increased GLUT4 expression [39]. It was also observed that this effect depends on the concentration of glucose that surrounds the cells, which suggests that the patter expression of GLUT4 observed in our study in the HIR-MO subjects could be due to the high concentration of circulating insulin observed, although there were no differences in circulating glucose levels. However, other studies observed contradictory results, showing an inhibitory effect of insulin on the level of GLUT4 mRNA [40]. 

Regarding LXR-α expression, we found an increased expression rate in the LIR-MO subjects in comparison with the HIR-MO subjects. Previous studies indicated that LXR-α is induced by insulin to regulate lipogenesis and lipid metabolism [41]. It has also been demonstrated that the activation of LXR-α impairs adipose expansion by increasing adipocyte apoptosis, lipolysis, and antagonizing the transcriptional activity mediated by PPAR-γ. This could contribute to a decrease in insulin sensitivity throughout the whole body [42]. These findings might explain what our study observed: that the LIR-MO individuals showed less expression of LXR-α in VAT in comparison with the HIR-MO individuals. Thus, it is possible that insulin resistance could proceed following other independent-LXR pathways or depend on other nuclear receptors.

The association between VAT, obesity, and insulin resistance is well known. Thus, the FA composition of VAT might reveal interesting information related to related-obesity disorders. In our study, we observed that the composition of FAs of VAT from the morbidly obese subjects was significantly different from that of the non-obese individuals. Indeed, a specific FA profile in VAT of morbidly obesity subjects was found. This profile includes high concentrations of palmitic acid (16:0), palmitoleic acid (16.1:n7), adrenic acid (22: 4n6), PUFA, omega 6 FA, and the SFA/PUFA ratio and low concentrations of arachidic acid (20:0). We also determined the specific FA profile for HIR-MO individuals, having a lower concentration of lauric acid (12:0), myristic acid (14:0), arachidic acid (20:0), DHA (22:6n3), and the omega 3/omega 6 ratio and increased concentration of eicosenoic acid (20:1n9), AA (20:4n6), and the AA/LA ratio.

The determination of the lipid profile could indicate what type of FA is used by the adipose tissue as a ligand/agonist to activate, regulate, or modulate specific gene expression in lipolysis or adipogenesis [43]. A study regarding VAT reported a distinct association pattern between lipid metabolizing genes and individual FAs [44]. Variations in the expression and activity of lipogenic enzymes, such as elongase and desaturase, may partly explain the specific differences of the FA deposits and composition [45]. However, there is a lack of knowledge about the associations between FAs and tissue-specific gene expressions. A complex relationship was found between FA composition and metabolism. The composition of FAs is not only regulated by endogenous processes, but also by diet. Indeed, FA species are ligands of nuclear receptors that directly modulate cellular processes through transcriptional regulation [43]. Therefore, the FA profile found in our study could provide solid clues to consider specific ligands, which not only activate lipogenesis as a mechanism against obesity, but also to know mechanisms to inhibit adipogenesis through their action on the nuclear receptors.

In our study, we found that AA was able to bind to PPAR-γ2 and further activate GLUT4 in vivo and in vitro. When we silenced the PPAR-γ2 gene, GLUT4 activation was decreased in the presence of AA compared with the negative control in the absence of siRNA–PPAR-γ2. This indicates that AA is an activator of GLUT4 gene expression via PPAR-γ2. This effect was observed by other studies that used the 3T3-L1 cell line but did not demonstrate the effect of PPAR-γ2 on the expression of GLUT4 [46]. The enhancement of glucose uptake depends, at least in part, on the activation of PPAR-γ2. Our results also suggest that the molecular mechanisms of AA and insulin that potentiate basal glucose uptake may differ slightly, showing an alternative insulin-independent glucose uptake. The systematic study of the effects of various FAs on glucose uptake in adipocytes was first reported by Grunfeld et al. [47]. This study reported an increased basal glucose uptake when 3T3-L1 cells were incubated in the presence of a wide range of SFAs and monounsaturated fatty acids (MUFAs) during the differentiation process. SFAs decreased insulin absorption, whereas MUFAs had no effect. Our experiments showed the effect of dietary components and the composition of adipose tissue or the cell membrane in another process independent of insulin for glucose uptake and insulin resistance. Our experiments showed the effect of dietary FAs on glucose metabolism by an insulin-independent process. Thus, these results could indicate one of the gene–diet interaction pathways, which could lead to new therapies against obesity and insulin resistance.

## 4. Materials and Methods 

### 4.1. Subjects

A total of 45 individuals were included in the present study. Among them were 11 non-obese subjects (body mass index (BMI) < 25 kg/m^2^) and 34 morbidly obese individuals (BMI > 40 BMI < 25 kg/m^2^), who were classified in 2 groups according to insulin resistance: 15 LIR-MO (HOMA-IR < 5) and 19 HIR-MO (HOMA-IR > 8) individuals. VAT biopsies from the obese patients and lean subjects were obtained by laparoscopic gastric bypass. The exclusion criteria were patients with type 2 diabetes mellitus, metabolic syndrome, coronary artery disease, inflammatory diseases such as arthritis and others or patients under treatment with drugs that could alter lipid and metabolic markers.

The samples from the subjects were processed and frozen immediately after their reception in the Regional University Hospital Biobank (Andalusian Public Health System Biobank). All the subjects (non-obese and morbidly obese subjects) were of Caucasian descent. All the participants gave their written informed consent, and the study was reviewed and approved by the Ethics and Research Committee of Regional University Hospital, Malaga, Spain.

### 4.2. Laboratory Measurements

The blood samples from all the subjects before surgery were collected after a 12-h fast. The serum was separated and immediately frozen at −80 °C. The serum biochemical variables were measured in duplicate. Serum glucose, cholesterol, high density lipoprotein (HDL) cholesterol, triglycerides, and FFAs were measured by standard enzymatic methods (Randox Laboratories Ltd., Antrium, UK). The adiponectin levels were measured by an enzyme immunoassay Adiponectin ELISA kit (DRG Diagnostics, Marburg, Germany). The leptin levels were measured by a Leptin ELISA kit (Mediagnost, Reutlingen, Germany). The insulin was analyzed by an immunoradiometric assay (BioSource International, Camarillo, CA, USA). The homeostasis model assessment of insulin resistance index (HOMA-IR) was calculated with the following equation: HOMA-IR = fasting insulin (μIU/mL) × fasting glucose (mg/dL)/405 [28]. The morbidly obese subjects were also divided in two groups as follows: those with low HOMA-IR (<4.7) and those with high HOMA-IR (>8).

### 4.3. Isolation of RNA and Quantitative RT-PCR

The total RNA isolation from VAT was extracted using a RNeasy lipid tissue mini kit (Qiagen GmbH, Hilden, Germany) following the manufacturer’s instructions. The purity of the RNA was assessed by measuring the 260/280 absorbance ratio. The integrity of the purified RNA was determined by denaturing agarose gel electrophoresis.

For the first strand of cDNA synthesis, constant amounts of 1 μg of total RNA were reverse transcribed using random hexamers as primers and M-MLV reverse transcriptase (Roche Diagnostic, Rotkreuz, Switzerland). The transcript levels for each gene were quantified by real-time reverse transcription (RT)-PCR, using LightCycler technology (Roche Diagnostic, Rotkreuz, Switzerland) with SYBR green detection. The following specific annealing primers (Sigma Proligo, Paris, France) for GLUT4 and LXR-α were designed using Primer-Blast (NCBI). A standard curve was created with serial dilutions of a PCR fragment from human adipose tissue total RNA (Clontech Laboratories, Inc., Mountain View, CA, USA). After this test, all the samples were quantified in at least two different runs. For quantification purposes, gene expression levels were always reported to the levels of the constitutively expressed gene β-actin. A threshold cycle (Ct value) was obtained for each amplification curve, and the Ct value of β-actin was left. The changes were determined by calculating 2^−ΔCt^. The results of the expression were represented as the target gene/β-actin ratio, according to the manufacturer’s guidelines. All the samples were quantified in triplicate, and positive and negative controls were included in all the reactions.

### 4.4. Western and Ligand Immunoblot Analysis 

Cytoplasmic and nuclear extracts were prepared from VAT of the non-obese and morbidly obese subjects using a NE-PER nuclear and cytoplasmic extraction reagent kit (Thermo Scientific, Rockford, IL) according to the manufacturer’s instructions. The protein concentration was determined using the Bradford method (Thermo Scientific, Rockford, IL, USA), using bovine serum albumin as a standard. The protein extracts (about 30 μg) were separated by SDS-PAGE, blotted onto a polyvinylidene difluoride (PVDF) membrane and blocked in Tris-buffered saline (TBS)-Tween 20 (50 mmol/L Tris-HCl (pH 7.5), 0.15 mol/L NaCl, and 0.1% Tween 20) containing 5% skimmed milk for 16 h at 4 °C. After washing, the membranes were incubated with specific antibodies, using mouse anti-LXR-α monoclonal antibody (clone PPZ0412, AbCAM), rabbit polyclonal anti-Glut4 (#07-1404, Millipore), and anti-β-actin (Part A1978, Sigma Aldrich, Madrid, Spain), overnight at room temperature. The membranes were washed and incubated with horseradish peroxidase-conjugated secondary antibody (Promega, Madison, WI, USA) for 1 h at room temperature. The protein signals were developed with Supersignal West Pico western blot detection kit (Thermo Scientific, Rockford, IL) and were detected by electrochemiluminescence detection Auto-Chemi system analysis software Labworks 4.6 (UVP; Bio-Imaging Systems). Densitometry analyses are presented as a relative ratio of the LXR-α/β-actin and GLUT4/β-actin ratio.

### 4.5. Visceral Adipose Tissue Fatty Acid Composition

FA composition was performed as described previously [48]. Total lipids from frozen VAT samples were extracted with chloroform-methanol 2:1 (*v*/*v*). FA methyl esters were prepared as published previously [49]. The samples were analyzed in a Hewlett-Packard 4890A gas chromatograph (Agilent Technologies, USA), equipped with a flame ionization detector and capillary column Supelco OMEGAWAX™ 320 (30 m × 0.32 mm × 0.25 μm film thickness) (Supelco, Bellefonte, PA). The oven temperature was maintained at 140 °C for the first few minutes and increased at a rate of 6 °C per minute until 240 °C was reached [50]. This temperature was maintained for 4 min. The identity of each FA peak was ascertained by a comparison of the peak’s retention time with the retention times of synthetic standards with known FA compositions.

### 4.6. Plasmids and DNA Constructs

#### 4.6.1. Protein Expression Vectors

Full-length cDNAs for human PPARγ-2, human retinoid X receptor alpha (RXR-α), and human SRC1 were subcloned into the T7/SV40 promoter-driven pSG5 expression vector (Stratagene-Agilent Technologies Inc., Santa Clara, CA, USA). The same constructs were used for both T7 RNA polymerase driven in vitro transcription/translation of the respective cDNAs and for viral promoter-driven overexpression of the respective proteins in mammalian cells.

#### 4.6.2. GST Fusion Protein Constructs

The nuclear receptor interaction region of SRC1 (spanning from amino acid 597 to 791) was subcloned into the pGEX glutathione S-transferase (GST) fusion vector (Amersham Biosciences-GE Healthcare Europe GmbH, Barcelona, Spain). This fragment was then subcloned into the *Bam*HI/*Eco*RI sites of the pGEX-2T plasmid.

#### 4.6.3. Reporter Gene Constructs

A total of 4 copies of the human GLUT4 gene type response element [GLUT4-RE] were individually fused with the thymidine kinase (*tk*) minimal promoter driving the firefly luciferase reporter gene.

#### 4.6.4. In Vitro Translation and Bacterial Overexpression of Proteins

A rabbit reticulocyte lysate system (Promega Co., Madison, WI, USA) was used for in vitro transcription/translation of the wild type PPARγ-2 and RXR-α, using [^35^S]-methionine as the label to quantify the protein batches by test translation. The individual number of methionine residue per protein was used to adjust the specific concentration of the receptor protein to 4 ng/μL. The overexpression of GST-SRC1_597–791_ or GST alone was conducted in *Escherichia coli* BL21 (DE3) pLysS strain (Stratagene-Agilent Technologies Inc., Santa Clara, CA, USA).

The overexpression was stimulated with 0.25 mM isopropyl-β-d-thiogalactopyranoside for 3 h at 37 °C, and the proteins were purified and immobilized on glutathione-Sepharose 4B beads (Amersham Biosciences-GE Healthcare Europe GmbH, Barcelona, Spain) according to the manufacturer’s protocol. The proteins were eluted in the presence of glutathione.

### 4.7. GST Pull-Down Assays

GST pull-down assays were performed with 50 μL of a 50% sepharose bead slurry of GST-Vector and GST-SRC1_597–791_ (preblocked with 1 mg/mL BSA) and 20 ng in vitro-translated, [^35^S]-labeled PPARγ-2 in the presence or absence of OA, LA, and AA (1 μM). The proteins were incubated in immunoprecipitation buffer [20 mM HEPES (pH 7.9), 200 mM KCl, 1 mM EDTA, 4 mM MgCl_2_, 1 mM DTT 0.1% Nonidet P-40, and 10% glycerol] for 20 min at 30 °C. In vitro translated proteins that were not bound to GST-fusion proteins were washed away with immunoprecipitation buffer. GST-fusion protein bound, [^35^S]-labeled PPARγ-2 were resolved by electrophoresis through 15% sodium dodecyl sulphate-polyacrylamide gels and quantified on a FLA-3000 reader (Fuji, Tokyo, Japan) using Image Gauge software (Fuji).

### 4.8. Transfection and Luciferase Assays

Human HepG2 (liver hepatocellular carcinoma cell line) cells were seeded into 6-well plates (200,000 cells/well) and grown overnight in phenol red-free Dulbecco’s modified Eagle’s medium (DMEM) supplemented with 5% charcoal-stripped fetal bovine serum. Cell transfection was performed using liposomes containing plasmid DNA, which were formed by incubating 1 μg of an expression vector for wild-type PPARγ-2, RXR-α and SRC1 (pSG5 vector) and 1 μg of reporter plasmid GLUT4 with 10 μg of N-[1-(2,3-Dioleoyloxy)]-N,N,N-trimethylammonium propane (DOTAP) from Roche Applied Science (Basel, Switzerland) for 15 min at room temperature in a total volume of 100 μL. After dilution with 900 μL phenol red-free Dulbecco’s modification of Eagle medium (DMEM), the liposomes were added to the cells. Phenol red-free DMEM supplemented with 500 μL of 15% charcoal-stripped fetal bovine serum was added 4 h after transfection. In all cases, the cells were treated for 16 h with solvent (DMSO) and 10 μM of the different compounds indicated. The cells were lysed 16 h after the onset of the stimulation using the reporter gene lysis buffer (Roche Applied Science, Basel, Switzerland). The constant light signal luciferase reporter gene assay was performed as recommended by the supplier. The luciferase activities were normalized with respect to the protein concentration. The stimulation of the normalized luciferase activity was calculated in comparison with solvent-induced cells that did not overexpress the protein.

### 4.9. Gene Silencing (siRNA)

Gene Silencing with small interfering RNA (siRNA) HepG2 (liver hepatocellular carcinoma cell line) (ATCC, HB-8065) cells were cultured following the manufacturer’s guidelines. Then, 2 sequences (D-003434-01, D-003434-02) for siRNA targeting human PPAR (Dharmaco Inc., Thermo Fisher Scientific, Lafayette, CO, USA) were used in the silencing assay according to manufacturer´s instructions. To test non-specific gene expression, we used the siRNA negative control from Dharmacon (D-001810-01) to assess transfection efficiency. The cells were serum deprived for 16 h before stimulation. Starved HepG2 cells were transfected using the transfection reagent Nucleofector kit and the Nucleofector II Device (Amaxa-Lonza Group Ltd., Basel, Switzerland) according to the manufacturer’s instructions. We used 6-well plates containing 100,000 cells/well and added 2.5 nM of D-003434-01 and 2.5 nM of D-003434-02 siRNAs per well. After 30 min, the cells were treated for 48 h with solvent as a negative control and rosiglitazone as a positive control and AA (10 μM; 0.5% serum).

### 4.10. Statistical Analysis

The results are given as the mean ± SD in tables and the mean ± SEM in figures. To determine the difference between the study groups, we used the Mann–Whitney test to compare 2 groups, the Kruskal–Wallis test to test more than 2 groups, and the Wilcoxon test to determine difference between related variables. Statistical analyses were performed using SPSS (Version 11.5 for Windows (SPSS, Chigaco, IL, USA)). *p* < 0.05 were considered to be statistically significant.

## 5. Conclusions

In summary, our results indicated a dysregulation of LXR-α and GLUT4 expression in VAT of morbidly obese subjects, which could help explain the possible mechanism of insulin resistance related to obesity. The FA composition found in this study may help to understand the role of different FAs in lipolysis and adipogenesis. Finally, AA was able to bind as a ligand of PPAR-γ2 and further activate GLUT4 expression, showing an alternative activation of GLUT4 in an insulin-independent manner. Thus, these results might lead to a novel pharmacological compound for the treatment of obesity and insulin resistance. 

## Figures and Tables

**Figure 1 ijms-20-00963-f001:**
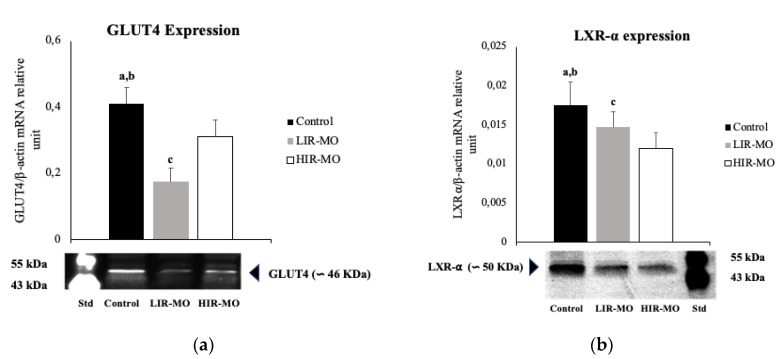
GLUT4 (**a**) and LXR-⍺ (**b**) mRNA expression and GLUT4 and LXR-⍺ protein levels in VAT from lean individuals and morbidly obese patients. The mRNA expression of GLUT4 and LXR-⍺ was normalized to β-actin levels. GLUT4 and LXR-⍺ mRNA expression levels in VAT in the controls (black bar) (*n* = 11), LIR-MO (grey bar) (*n* = 15), and HIR-OR groups (white bar) (*n* = 19). The results are given as the mean relative mRNA expression ± standard deviation. Different letters indicate significant differences between the means of the different groups of subjects (*p* < 0.05; a: controls vs. LIR-MO; b: controls vs. HIR-MO; c: LIR-MO vs. HIR-MO) according to Student’s *t*-test. Abbreviations: GLUT4: glucose transporter type 4; LXR-⍺: liver X receptor type alpha; LIR-MO: low insulin resistance morbidly obese; HIR-MO: high insulin resistance morbidly obese; and KDa: kilodalton.

**Figure 2 ijms-20-00963-f002:**
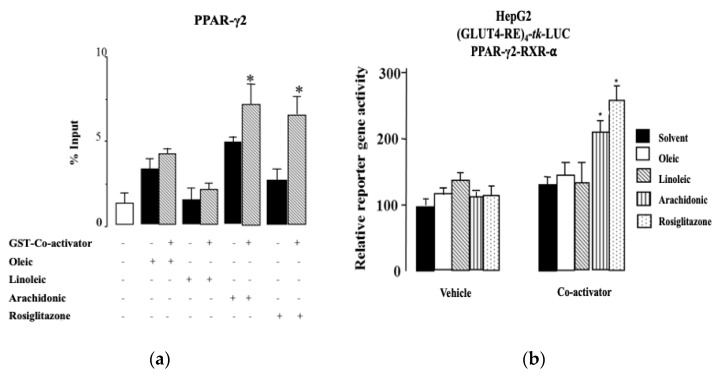
Ligand-dependent interaction profiles between the PPAR-γ2 and co-activators on DNA. (**a**) In vitro experiment using GST-pull down assays with GST-SRC1597-791 (Co-activator) and [^35^S]-labeled wild-type PPAR-γ2 in the presence of the vehicle (DMSO) (1 µM), 1 µM OA, LA, and AA. As a positive control, we used rosiglitazone (1 µM). The percentage of precipitated PPAR-γ2 proteins with respect to the input was quantified. The columns represent means ± SEM of at least three experiments, and the data were analyzed by Student’s t test (* *p* < 0.05). Significative differences were denoted compared with the GST-SRC1 interaction in the presence of the ligands. (**b**) In vivo experiment using luciferase reporter gene assays to determine the basal and ligand-induced activity of PPARγ-2. HepG2 cells were transiently transfected with a reporter construct containing human (GLUT4-RE)_4_-tk-LUC and the indicated expression vectors for PPARγ-2, RXRα and SRC1 (co-activator). HepG2 cells were treated for 16 h with 10 μM of the vehicle (DMSO), OA, LA, and AA as rosiglitazone (positive control) for PPARγ-2. The stimulation of the luciferase activity was normalized to the basal activity of PPARγ-2–RXRα in the absence of co-activators. The percentage of precipitated PPAR-γ2 proteins with respect to the input was quantified. The columns represent means ± SEM of at least three experiments, and the data were analyzed by Student’s *t* test (* *p* < 0.05). Significant differences were denoted compared with the GST-SRC1 interaction in the presence and absence of ligands. Abbreviations: PPAR-γ2: peroxisome proliferator-activated receptor gamma type 2; GST: glutathione S-transferase; GST-co-activator or GST-SRC1: glutathione S-transferase-tagged steroid receptor coactivator 1-receptor interaction domain 597 791; DMSO: dimethyl sulfoxide; GLUT4: glucose transporter type 4; HepG2: liver hepatocellular carcinoma cell line; (GLUT4-RE)_4_-tk-LUC: GLUT4 reporter fused with thymidine kinase luciferase activity; RXR: retinoid X receptor; OA: oleic acid; LA: linoleic acid; and AA: arachidonic acid.

**Figure 3 ijms-20-00963-f003:**
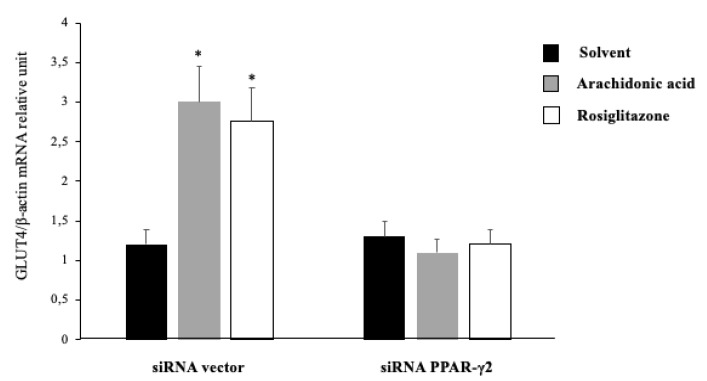
GLUT4 mRNA expression in HepG2 cells after PPAR-γ2 gene silencing by siRNA. The mRNA expression of GLUT4 in HepG2 cells were determined by qRT-PCR, which were serum deprived for 16 h and transfected with PPARγ-2-specific siRNA or control siRNA, followed by incubation with 10 μM of solvent (DMSO) as a negative control, AA, and rosiglitazone as positive control for 48 h. The expression was normalized to β-actin. The columns represent means ± SEM of at least three experiments, and the data were analyzed by Student’s t test (* *p* < 0.05 denote significant differences compared with the vehicle). Abbreviations: GLUT4: glucose transporter type 4; PPARγ-2: peroxisome proliferator-activated receptor gamma type 2; siRNA: small interference RNA; HepG2: liver hepatocellular carcinoma cell line; DMSO: dimethyl sulfoxide; and AA: arachidonic acid.

**Table 1 ijms-20-00963-t001:** Baseline anthropometric and biochemical variables of the study population.

Variables	Control (*n* = 11)	LIR-MO (*n* = 15)	HIR-MO (*n* = 19)
Sex (male/female)	11 (4/7)	15 (5/10)	19 (7/12)
Age (years)	44.6 ± 10.8	43.1 ± 11.2	41.1 ± 7.6
Weight (kg)	71.1 ± 12.6 ^a,b^	145.3 ± 30.7	156.4 ± 25.1
BMI (kg/m^2^)	23.0 ± 1.6 ^a,b^	53.8 ± 6.9	56.4 ± 6.8
Waist circumference (cm)	86.3 ± 4.0 ^a,b^	136.9 ± 19.7	146.8 ± 18.9
Glucose (mg/dL)	88.0 ± 5.7 ^b^	94.6 ± 9.7 ^c^	110.6 ± 17.6
Insulin (µIU/mL)	10.7 ± 3.1 ^a,b^	17.1 ± 4.6 ^c^	38.6 ± 14.5
Total cholesterol (mg/dL)	178.0 ± 44.1	195.9 ± 44.8	196.1 ± 28.6
FFA (mmol/L)	0.278 ± 0.133 ^a,b^	0.635 ± 0.323	0.663 ± 0.241
Triglycerides (mg/dl)	72.9 ± 37.0 ^a,b^	111.4 ± 46.1 ^c^	189.6 ± 79.9
Leptin (ng/mL)	11.8 ± 2.7 ^a,b^	156.2 ± 90.9	149.6± 87.6
Adiponectin (ng/mL)	18.2 ± 5.9 ^a,b^	11.4 ± 4.0	8.5 ± 4.3
HOMA-IR	2.81 ± 1.22 ^a,b^	3.96 ± 1.08 ^c^	10.37 ± 5.07

The results are given as the mean ± standard deviation. Different letters indicate significant differences between the means of the different groups of subjects (*p* < 0.05; a: controls vs. LIR-MO; b: controls vs. HIR-MO; c: LIR-MO vs. HIR-MO) according to Student’s *t*-test. Abbreviations: BMI: body mass index; HOMA-IR: homeostasis model assessment of insulin resistance; LIR-MO: low insulin resistance morbidly obese; HIR-MO: high insulin resistance morbidly obese; and FFA: free fatty acids.

**Table 2 ijms-20-00963-t002:** Fatty acid composition in the visceral adipose tissue of normal-weight and obese subjects.

Fatty Acids	Control (*n* = 11)	LIR-MO (*n* = 15)	HIR-MO (*n* = 19)
12:0 (lauric)	0.37 ± 0.20 ^b^	0.46 ± 0.31 ^c^	0.21 ± 0.11
14:0 (myristic)	1.75 ± 0.55 ^b^	1.91 ± 0.45 ^c^	1.36 ± 0.33
16:0 (palmitic)	16.61 ± 2.13 ^a,b^	18.78 ± 2.58	18.51 ± 1.70
16.1:n7 (palmitoleic)	3.21 ± 0.78 ^a,b^	4.48 ± 1.24	4.02 ± 0.88
18:0 (stearic)	3.94 ± 1.26 ^b^	3.11 ± 1.01	2.79 ± 0.69
18:1n9 (oleic)	51.64 ± 8.07	48.93 ± 5.18	50.33 ± 5.24
18:2n6 (linoleic)	18.37 ± 6.18	17.83 ± 2.14	18.26 ± 4.38
18:3n6 (α-linolenic)	0.43 ± 0.06 ^b^	0.33 ± 0.16	0.38 ± 0.06
18:3n3 (γ-linolenic)	0.016 ± 0.0188 ^b^	0.036 ± 0.034	0.037 ± 0.025
20:0 (arachidic)	0.24 ± 0.03 ^a,b^	0.11 ± 0.012 ^c^	0.14 ± 0.03
20:1n9 (eicosenoic)	0.72 ± 0.18	0.67 ± 0.06 ^c^	0.75 ± 0.14
20:4n6 (arachidonic)	0.25 ± 0.05 ^b^	0.28 ± 0.20^c^	0.41 ± 0.06
20:5n3 (eicosatetraenoic)	0.040 ± 0.031	0.045 ± 0.020	0.043 ± 0.010
22:4n6 (adrenic)	0.14 ± 0.04 ^a,b^	0.24 ± 0.10	0.25 ± 0.08
22:5n3 (docosapentaenoic)	0.15 ± 0.07	0.18 ± 0.08	0.16 ± 0.05
22:6n3 (docosahexaenoic)	0.30 ± 0.08 ^b^	0.25 ± 0.06 ^c^	0.20 ± 0.04
SFA	25.47 ± 3.50 ^b^	24.39 ± 3.28	23.05 ± 2.42
MUFA	59.62 ± 4.28	56.14 ± 5.09	57.03 ± 5.68
PUFA	14.90 ± 1.17 ^a,b^	19.45 ± 2.25	19.91 ± 4.48
n3 (Omega 3 fatty acid)	0.51 ± 0.20	0.52 ± 0.14	0.45 ± 0.11
n6 (Omega 6 fatty acid)	14.38 ± 1.22 ^a,b^	18.93 ± 2.31	19.45 ± 4.39
n3/n6	0.036 ± 0.016 ^b^	0.028 ± 0.010 ^c^	0.023 ± 0.003
SFA/MUFA	0.43 ± 0.08	0.44 ± 0.09	0.40 ± 0.07
SFA/PUFA	1.70 ± 0.19 ^a,b^	1.25 ± 0.13	1.19 ± 0.24
Arachidonic/linoleic	0.016 ± 0.008 ^b^	0.015 ± 0.010 ^c^	0.023 ± 0.006
Docosapentaenoic/arachidonic	0.15 ± 0.10	0.11 ± 0.05	0.10 ± 0.03

The results are given as the mean ± standard deviation. Different letters indicate significant differences between the means of the different groups of subjects (*p* < 0.05; a: controls vs. LIR-MO; b: controls vs. HIR-MO; c: LIR-MO vs. HIR-MO) according to Student’s *t*-test. Abbreviations: LIR-MO: low insulin resistance morbidly obese; HIR-MO: high insulin resistance morbidly obese; SFA: saturated fatty acid; MUFA: monounsaturated fatty acid; and PUFA: polyunsaturated fatty acid.

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
