# Peer review of "The Antagonist Effect of Arachidonic Acid on *GLUT4* Gene Expression by Nuclear Receptor Type II Regulation"

_ijms, 2019, doi:10.3390/ijms20040963_

Reviewer 1 Report

In the present study, authors determined the decreased levels of GLUT4 and LXRa in adipose tissue obtained from non-obese, obese with low IR, and obese with hgih IR. In addition, altered fatty acid composition was also determined. Futhermore, using cell-based and in vitro assay, authors demonstrated that arachidonic acid (AA) activates PPARg2 to induce GLUT4 gene. Taken all together, authors presented an idea that arachidonic acid binds to and activates PPARg2 to induce GLUT4 in adipose tissue. Some part of their study looks interesting, but there are many flaws regarding line of flow, experiment and word usage.

Major comments

choice of lioleic acid and oleic acid as negative control for cell base and in vitro experiments

If AA controls incrased expression of GLUT4 in high IR obese subjects compared with low IR obese subjects, the lipid increased in obese compared with non-obese should also be used. For example, palmitate, palmitoleate, adrenate.

lacking control

Cell-based experiment and in vitro experiment lacks their controls and that makes obsecure what the expeirment indicate. For example, Fig. 2B lacks data without PPARg2/RXRa. As author demonstrated Fig. 3, PPARg2 should be expressed in HepG2 cells to activate GLUT4 gene. In addition, Fig 3 lacks the data obtained siRNA control with treatment.In addition, author should demonstrate that siRNA-PPARg2 effectively knockdown the endogenous PPAR2g.

AA concentration

The concentration of AA used this study is inconsistent. Authors used 1 uM for binding assay (Fig 2A), 10 mM for LUC experiment (Fig 2B), and 0.1 mM for siRNA experiment (Fig 3). Especially, as far as my experience, 10 mM of fatty acid usually cause apopotsis and not suitable for treatment.

Overall statement

According to the restuls, there is complex mechansim of the GLUT4 gene regulation. PPARg2 and LXR both regulates GLUT4 and their contribution looks different in obese with low IR and obese with high IR. I can understand the difficulty, however, authors should make more effort to let readers to understand easily.

Minor comments

1. WB data in Fig 1A and 1B

Author should not use color arrangement for figures.

2. The word lipidomics

Lipidomics ussually indicate overviwing lipid profiles including phosphoglycerolipids, sphingolipids, glycerolipids, and other lipids. A word "Lipidomics" should not be used for determination of fatty acid composition.

Author Response

REVIEWER 1

Choice of linoleic acid and oleic acid as negative control for cell base and in vitro experiments. If AA controls increased expression of GLUT4 in high IR obese subjects compared with low IR obese subjects, the lipid increased in obese compared with non-obese should also be used. For example, palmitate, palmitoleate, adrenate.

-Firstable, we do not use linoleic and oleic acids as negative controls in the figure2 y 3, we used them to analyze if this AAs could act in the interaction of PPARg2 in the regulation of Glut4, because they are abundantly present in the Mediterranean diet.

One of the common components and basic components in the characterization of the Mediterranean diet, is olive oil, traditionally used for cooking and food preparation. The olive oil is formed up to 75% oleic acid. The average consumption of olive oil in Spain is approximately 33 g / day (approximately 12 kg per person per year), which represents 60% of the total consumption of oils and fats. This consumption is somewhat lower than that of Greece (20 kg per person per year), but higher than that of France (400 g per person per year). The use of olive oil as the main source of lipids in the diet has characterized the countries of the Mediterranean region against the countries of northern Europe, which basically use animal-type fats.

In the other side, the Mediterranean diet is also rich in omega 3 and omega 6, in particular, linoleic acid. It is the main representative of the family of omega 6 fatty acids. The foods typically consumed in the Mediterranean diet contain a high content of linoleic acid, so we consider of great interest from the molecular point of view. From linoleic acid, our body produces other omega 6 fatty acids, such as arachidonic acid. Although the diet could provide sufficient arachidonic acid, this compound becomes essential if there is a deficiency in linoleic acid or if there is an inability to convert linoleic acid to arachidonic acid.

For these reasons, all patients included in this study follow the Mediterranean diet, we consider the interest to include oleic, linoleic and arachidonic acid, since it could give us a broader view of the interaction between gene-diet.

However, we understand the comment of the reviewer after to see the differences in the composition of other FAs, such as palmitoleic acid, between normal weight and obese subjects in the Table 2. In this aspect, we made different in vitro experiments with palmitic and palmitoleic acids, but we do not found any activation with PPARg2 in cell culture and in vitro experiments. We introduce this comment in the results section 2.4. Arachidonic acid binds to PPAR-g2 and their ligands actions, paragraphs 280-282, such as data not shown.

Comment 2:

lacking control

Cell-based experiment and in vitro experiment lacks their controls and that makes obsecure what the experiment indicate. For example, Fig. 2B lacks data without PPAR-g2 /RXRa. As author demonstrated Fig. 3, PPAR-g2 should be expressed in HepG2 cells to activate GLUT4 gene. In addition, Fig 3 lacks the data obtained siRNA control with treatment. In addition, author should demonstrate that siRNA- PPARg2 effectively knockdown the endogenous PPAR2g.

Answer 2:

We appreciate the comments of the reviewer, we spend more time to answer this part and we consider that help us so much to clarify the figures of the manuscript. Concerning to the lacking control without PPARg2/RXRa (Figure 2B), we would like to explain that the aim of this Reporter Gene assay was to verify which ligands are agonist of the heterodimer PPARg2/RXRa binding to the promoter of Glut4, for that we overexpressed those nuclear receptors with in vitro transcription/translation of the wild type PPARg-2 and RXR-α in presence of human GLUT4 gene type response element [GLUT4-RE] were individually fused with the thymidine kinase (tk) minimal promoter driving the firefly luciferase reporter gene. (See section methods). In this in vitro experiments we measured the activities of the luciferase according with the presence of different ligands in presence of coactivators to evaluate if they are agonists or not. Our negative controls used are in absence of coactivator and the ligand solvent to check the specificity of these protein-protein interactions. We also used a positive control, a synthetic agonist of PPARg2 rosiglitazone to check that our in vitro system is working, demonstrating that AA is an agonist of this heterodimer PPARg2-RXRa binding specifically to the promotor sequences of Glut4 in the transfection and luciferase assays.

Related with the Figure 3, we are totally agree with the reviewer and we accepted that the figure need to be revised to avoid the confusion of controls and ligands

We made a new figure, to see if the interaction of AA with PPARg2 change the mRNA expression of Glut4, so we transfected in the human cell lines with the vector of the siRNA (control) and with siRNA PPARg2, we could observed that the expression of Glut4 increased with rosiglitazone (agonist of PPARg2) and AA, but this effect we can not see when we silencing PPARg2 of the cell system. The reviewer need to consider that the level of expression of mRNA Glut4 is enough to see the diffents but the values are not so high (Figure 2), because the cells are working with their endogenous concentration of PPARg2 to see the physiological effect without overexpression of the system like in the Figure 2B

Comment 3:

AA concentration

The concentration of AA used this study is inconsistent. Authors used 1 uM for binding assay (Fig 2A), 10 mM for LUC experiment (Fig 2B), and 0.1 mM for siRNA experiment (Fig 3). Especially, as far as my experience, 10 mM of fatty acid usually cause apopotsis and not suitable for treatment.

We are totally agree with the reviewer, we already corrected in the text considering the right concentrations of the AA: 1uM for binding assays (fig 2A)  and 10 uM for the in vivo assays (figure 2b and 3). We apologize because 10 mM was a big mistake in the in vivo experiments as suggest the reviewer

Comment 4:

Overall statement

According to the results, there is complex mechanism of the GLUT4 gene regulation. PPARg2 and LXR both regulate GLUT4 and their contribution looks different in obese with low IR and obese with high IR. I can understand the difficulty, however, authors should make more effort to let readers to understand easily.

We appreciate this comment and we worked very hard in this aspect checking the grammar and all the expressions in the Introduction, results and discussion for a better understanding of the manuscript. We also revised the methods section to simplify and avoid any overlap compared

Comment 5:

Minor comments

1. WB data in Fig 1A and 1B

Author should not use color arrangement for figures.

We already do not see any color in the figures

2. The word lipidomics

Lipidomics ussually indicate overviwing lipid profiles including phosphoglycerolipids, sphingolipids, glycerolipids, and other lipids. A word "Lipidomics" should not be used for determination of fatty acid composition.

we already replaced in all the manuscript the word “lipidomics” for fatty acid profile according to the comment of the reviewer

Reviewer 2 Report

The manuscript deals with investigation of the antagonistic effects of the AA on Glut4 expression by the nuclear receptor II regulation. I have following comments: 

 1. Fig 1: Loading controls from PCR and Western blot data is required to be included. Most importantly, although the Y-axis says its relative expression, would authors state what those bars are normalized to? The control bar diagram does not seem to be at 1.0, rather its at 0.4? Same with fig b as well. Also, authors need to run at least 3 samples from each experimental group. Also the blots seems to be overprocessed. 

 2. There are several grammatical mistakes and the sentences are difficult to understand. I’d recommend authors to submit manuscript to native English speaker to enhance the enthusiasm of the paper. Below are a few egs.:

 Line 78: “..have been evaluated” to “have shown”?. Please revise the whole sentence, its difficult to understand.

 Line 80: ..was found? –remove “was”. 

 Line 87: “considering in”? Revise the sentence.

 Line 116: ppar-g2?

Author Response

REVIEWER 2

Fig 1: Loading controls from PCR and Western blot data is required to be included. Most importantly, although the Y-axis says its relative expression, would authors state what those bars are normalized to? The control bar diagram does not seem to be at 1.0, rather its at 0.4?. Same with fig b as well. Also, authors need to run at least 3 samples from each experimental group. Also the blots seems to be overprocessed.

We changes the title in the Y-axis and explained in the legends and also included in the section methods, related to the figure 1a, 1b and 3  that the relative expression is not normalized to fold change 1, it is made with the b-actin values after to consider the normalization following the method 2∆∆Ct   

We also included the molecular weight of the standards used in western blotting assays, and all the experiments are made in triplicate and is indicated in the text

Comment 2:

2. There are several grammatical mistakes and the sentences are difficult to understand. I’d recommend authors to submit manuscript to native English speaker to enhance the enthusiasm of the paper. Below are a few egs.:

Line 78: “..have been evaluated” to “have shown”?. Please revise the whole sentence, its difficult to understand.

CHANGED IN THE TEXT

 Line 80: ..was found? –remove “was”.

CHANGEN IN THE TEXT

 Line 87: “considering in”? Revise the sentence.

REVISED

 Line 116: ppar-g2?

CHANGED

Round  2

Reviewer 1 Report

The manuscript was improved.

Reviewer 2 Report

N/A